# Nutrient Supply Is Essential for Shifting Tree Peony Reflowering Ahead in Autumn and Sugar Signaling Is Involved

**DOI:** 10.3390/ijms23147703

**Published:** 2022-07-12

**Authors:** Yuqian Xue, Jingqi Xue, Xiuxia Ren, Changyue Li, Kairong Sun, Litao Cui, Yingmin Lyu, Xiuxin Zhang

**Affiliations:** 1Beijing Key Laboratory of Ornamental Germplasm Innovation and Molecular Breeding, China National Engineering Research Center for Floriculture, College of Landscape Architecture, Beijing Forestry University, Beijing 100083, China; xueyuqian2046@163.com; 2Key Laboratory of Biology and Genetic Improvement of Horticultural Crops, Ministry of Agriculture and Rural Affairs, Institute of Vegetables and Flowers, Chinese Academy of Agricultural Sciences, Beijing 100081, China; xuejingqi@caas.cn (J.X.); renxiuxia@caas.cn (X.R.); 15595662810@163.com (C.L.); skrskr@foxmail.com (K.S.); cuilitaoacca@163.com (L.C.)

**Keywords:** *Paeonia suffruticosa*, forcing culture, nutrient supply, sugar signal, gene expression

## Abstract

The flowering time of tree peony is short and concentrated in spring, which limits the development of its industry. We previously achieved tree peony reflowering in autumn. Here, we further shifted its reflowering time ahead through proper gibberellin (GA) treatment plus nutrient supply. GA treatment alone initiated bud differentiation, but it aborted later, whereas GA plus nutrient (G + N) treatment completed the opening process 38 days before the control group. Through microstructural observation of bud differentiation and starch grains, we concluded that GA plays a triggering role in flowering induction, whereas the nutriment supply ensured the continuous developing for final opening, and both are necessary. We further determined the expression of five floral induction pathway genes and found that *PsSOC1* and *PsLFY* probably played key integral roles in flowering induction and nutrient supply, respectively. Considering the GA signaling, *PsGA2ox* may be mainly involved in GA regulation, whereas *PsGAI* may regulate further flower formation after nutrient application. Furthermore, G + N treatment, but not GA alone, inhibited the expression of *PsTPS1*, a key restricting enzyme in sugar signaling, at the early stage, indicating that sugar signaling is also involved in this process; in addition, GA treatment induced high expression of *PsSnRK1*, a major nutrient insufficiency indicator, and the induction of *PsHXK1*, a rate-limiting enzyme for synthesis of sugar signaling substances, further confirmed the nutrient shortage. In short, besides GA application, exogenous nutrient supply is essential to shift tree peony reflowering ahead in autumn under current forcing culture technologies.

## 1. Introduction

Tree peony (*Paeonia* sect. *Moutan*) is a traditional woody plant and the candidate national flower of China, which is generally attracting people worldwide because of its abundant flower types and gorgeous colors. All nine wild-tree peony species are native to China, with ornamental, cultural, and economic value. Take Heze, one of the main tree peony producing cities in China as an example, its total output value of tree peony industry exceeded CNY 9 billion (about USD 1.4 billion) in 2020 [1]. However, the short flowering period and single-season flowering seriously limited its current industrialized production. Prolonging the flowering period or achieving multi-season flowering is a hot spot in tree peony research. In our previous studies, we achieved tree peony reflowering in autumn and winter through variety selection and forcing cultivation improvements [2,3]. Based on these works, most tree peony cultivars showed good reflowering performance in winter, but the reflowering in autumn still showed strict variety specificity, and further shifting the reflowering time is also very difficult [2,4]. The full differentiation and formation of flower buds in plants is the key step for flowering, and determining this process in tree peony will be helpful for understanding its reflowering mechanism.

In order to reveal the molecular mechanism of the flowering process in higher plants, scientists isolated a large number of early and late-flowering mutants from Arabidopsis and some other plants, and identified a serious of flowering-regulation genes. Based on these studies, the overall flowering pathways can be classified into three categories: photoperiod pathways, including light quality, light intensity and circadian clock; plant internal signaling pathways, including plant hormones, such as gibberellins (GAs), nutrients, such as sugars, nitrogenous compounds, etc., and aging; other pathways, including autonomic, vernalization, and ambient temperature pathways [5,6,7,8,9,10]. Among them, *FLOWERING LOCUS T* (*FT*), *SUPPERSSOR OF CONSTANS OF OVEREXPRESSION1* (*SOC1*), and *LEAFY* (*LFY*) have been proven to be integration factors in various flowering pathways [11]. These genes were also confirmed to be involved in the regulation of tree peony reflowering in autumn according to our previous work [12]. Furthermore, *APETALA1* (*AP1*), which functions downstream in the floral regulation pathway, regulates both the transformation of inflorescence meristems to floral meristems and the morphological development of floral organs [13], which is activated by the combined action of *FT* and *LFY* in Arabidopsis [14]. Besides positive regulation, some genes are also reported to inhibit floral formation, such as *SHORT VEGETATIVE PHASE* (*SVP*), which plays an important role in the ambient temperature flower-forming pathway for the vegetative growth phase [15].

Flowering is adaptable to environmental conditions, and it is completed under a complex floral regulation network formed by a variety of exogenous and endogenous signals. As one of the most important endogenous signals, plant hormones, including GA, play an important role in the process of flower formation. The GA pathway is one of the earliest-discovered flower formation pathways, and with regarding to transcriptional level, *GA INSENSITIVE DWARF 1* (*GID1*), *GA20 Oxidase* (*GA20ox*), *GA3 Oxidase* (*GA3ox*), *GA2 Oxidase* (*GA2ox*), and *GA INSENSITIVE* (*GAI*) are considered as major regulatory factors [16]. In rice, GID1 is the only GA receptor discovered to date. When the self-synthesized endogenous GAs or exogenous GAs are combined with GID1, they can jointly degrade the DELLA protein through the ubiquitination pathway, thereby regulating downstream genes [17,18]. As one of the *DELLA* genes, *GAI* mediates the transduction of the GA signal [19]. The pathways of GA biosynthesis and metabolism are catalyzed by three enzymes, namely, GA20ox, GA3ox, and GA2ox [16]. Under the regulation of these three enzymes, the dynamic balance of active gibberellin is well maintained.

In addition to plant hormones, sugar is an important signal compound in many plants. In the sugar signal pathway, trehalose-6-phosphate (T6P) is considered as the most critical signal molecules in the process of plant growth and development, participating in flower induction and floral organ development, and trehalose-6-phosphate synthase (TPS) is a key enzyme that restricts T6P biosynthesis [20]. Sucrose non-fermenting-1-related kinase 1 (SnRK1), another key signal component in the sugar signal pathway, responds to changes in nutrient levels in many plants, such as Arabidopsis [21], tomato [22] as well as tree peony, as we reported previously [3]. SnRK1 is active at low nutrient levels, and mainly inhibits the biosynthesis process and plant growth [23]. Hexokinase1 (HXK1), as another key factor in the sugar signal pathway, converts glucose into glucose-6 phosphate and participates in the process of intracellular sugar signal transduction [24].

Tree peony ‘Qiu Fa No. 1’ is one of our previously selected cultivars with good autumn-reflowering characteristics, and GA treatment induced its bud development in autumn, but the bud failed to bloom [2]. We speculated this may be due to nutritional deficiency; thus, in this study, we used tree peony ‘Qiu Fa No. 1’ as the material and treated it with GA and a water-soluble fertilizer as the nutrient supply in autumn, but earlier than before (detailed in Materials and Methods), to further shift the flowering time under current forcing culture conditions [4]. We observed the morphological and microstructural changes in buds, determined GA_3_ content and the expression of genes related to floral induction, GA signaling, and the sugar signal pathway. This work may provide a theoretical reference for improving current tree peony forcing culture technology to achieve annual flowering with better quality.

## 2. Results

### 2.1. Effects of Different Treatments on the Morphological Changes in Tree Peony Flower Buds during Growth and Development

In this study, we first determined the monthly environmental parameters in the nursery during the experimental period, including temperature, light, and humidity (Appendix A). Then, we recorded the effects of different treatments on the morphological changes in flower buds from 0 to 90 d. The results showed that in the control group, the flower buds increased slowly from 0 to 60 d and formed a hard bud at 90 d. In the GA treatment group, the flower buds were significantly enlarged at 30 d, especially the horizontal diameter, and germinated at 45 d. The buds developed into small-sized budlets at 60 d but eventually aborted at 90 d. In the G + N treatment group, the bud development showed a similar trend to the GA treatment group during the first 45 d, continuously enlarged until 60 d, and finally flowered at 90 d, which was 38 d earlier than the control with the flowering rate of 62.6% (Figure 1 and Appendix A).

We then observed the internal microstructure of buds through paraffin sections. The results showed that in the control group, the buds underwent continuous growth and development from 0 to 60 d, from undifferentiated to the bract primordium stage. After GA treatment, the bud growth accelerated, and reached the sepal primordial stage at 60 d. The G + N treatment also promoted bud differentiation, the buds developed into pistil primordial stage at 60 d, and the entire bud differentiation was completed (Figure 2A). We also statistically analyzed the ratio of the transverse to the longitudinal diameter of the buds. In the control group, this ratio decreased consistently, from 2.3 at 0 d to 1.1 at 60 d; in both the GA and G + N groups, this decrease was significantly slower by 60 d, with average ratios of 1.8 and 1.6, respectively (Figure 2B).

### 2.2. Effects of Different Treatments on the Starch Metabolism in Tree Peony Buds

In our previous study, we found that under the forcing culture conditions in autumn, flower buds need to continuously consume starch stored in cells during the dormancy release process [4]. In the present study, the starch granule number significantly increased in all three groups at 30 d, with the higher numbers in the GA and G + N treatment groups; after that, the starch granule number in both the GA and G + N treatment groups decreased at 60 d, especially in the GA treatment group, whereas in the G + N group, this number was still greater than that in the control group (Figure 3A,B). The size of the starch granules increased 3.1 times in the control group at 30 d and remained constant at 60 d; this increase was delayed in both the GA and G + N treatment groups, and in the G + N group, the starch granule diameter sharply increased to 76.3 μm, which far exceeded that of the other two groups (Figure 3C).

### 2.3. Effects of Different Treatments on Soluble Sugar Content in Tree Peony Buds

The sucrose content in the control group increased at 15 d, and then declined after that; in the GA treatment group, its level decreased at 15 d, and then slight recovered; the G + N group showed a similar trend to that of the control group, with the lower level at 30 d (Figure 4A). The glucose and fructose content showed a similar trend: in the control group, their content remained relatively stable in the first 45 d, and had a slight drop after that; GA treatment dramatically induced their content at 30 d, whereas in the G + N group, this inducing advanced their content to 15 d, with a higher level (Figure 4B,C).

### 2.4. Effects of Different Treatments on the Expression of Related Genes in Tree Peony Flower Buds during Growth and Development

#### 2.4.1. Floral Pathway-Related Genes

In this study, the expression of *PsFT* in the control group was relatively low in the first 30 d, then significantly increased from 45 d onward. This increase was dramatically enhanced in the GA treatment at 45 d, at 13.5 times that of the control group. In the G + N treatment, this increase was advanced to 30 d, although the expression of *PsFT* was lower than that in the GA treatment group at 45 and 60 d (Figure 5A). The expression of *PsSOC1* showed a single-peak trend in the control and peaked at 45 d; both the GA and G + N treatment groups advanced this peak to 15 d, and this level was maintained to 30 d in the G + N group (Figure 5B). In contrast, the expression of *PsLFY* continuously increased in all three groups; its expression in the GA treatment group was lower than that in the control group at all the sampled time points, and its level in the G + N group was greater than that in the control from 30 to 60 d (Figure 5C). The expression of *PsAP1* slightly increased during the bud growth process, with an obvious increase at 60 d. The GA treatment significantly increased the level of *PsAP1* at both 45 and 60 d, especially at 60 d, in which expression was 49.5 times that of the expression at 0 d (Figure 5D). The expression of *PsSVP* exhibited a single-peak curve in the control group, which reached the maximum at 45 d; both the GA and G + N treatment groups had a lower level at 45 d, and its level continued to increase in the G + N group, which surpassed the control group at 60 d (Figure 5E).

#### 2.4.2. GA Signal Pathway-Related Genes

In order to clarify the effects of different treatments on the expression of GA signal genes, we first measured the GA_3_ content at 15 d and found that GA treatment and G + N increased it by 3.76 and 1.98 times, respectively (Figure 6A). We then determined the expression of five related genes. As one of the key genes for bioactive GA synthesis, *PsGA20ox* expression remained low in the control group; in GA treatment group, its expression was significantly induced from 45 d and increased to an extremely high level at 60 d. Similarly, its expression was also induced by G + N, but much earlier from 15 d (Figure 6B). For *PsGA3ox*, its expression pattern was similar to that of *PsGA20ox*, whereas the expression in G + N group induced much greater at 15 d (Figure 6C). The expression of *PsGA2ox* in the control group was also low throughout the process, while the GA and G + N treatments significantly increased its expression at 15 d by 24.4 and 83.3 times, respectively, and the level in the G + N group began to fall, but it remained greater than that of the control until the end of the experiment (Figure 6D). The expression of *PsGAI* showed a similar trend in all three groups, with the expression level in the GA treatment group greater than that of the other two groups at 30 and 45 d (Figure 6E). The expression of *PsGID1c* increased from 30 to 45 d in the control group. Both the GA and G + N treatments inhibited this increase, and the expression remained at a similar level during all the sampling periods in both groups (Figure 6F).

#### 2.4.3. Sugar Signal Pathway-Related Genes

Sugar has been confirmed as a new signal substance in recent years. In this study, we also determined the expression of three signal pathway-related genes. The results showed that the expression of *PsTPS1* increased in all three groups from 15 d; this increase began to decline at 45 d and 60 d in the control and GA treatment groups, respectively, whereas no decline occurred in the G + N group (Figure 7A). The expression of *PsSnRK1* showed a single-peak curve, and the GA and G + N treatments increased its expression from 15 d and 30 d, respectively (Figure 7B). The expression of *PsHXK1* was stable in the control group, whereas both the GA and G + N treatments significantly increased its expression with similar levels at 15 d, and in the G + N group, its expression rapidly declined after that (Figure 7C).

## 3. Discussion

Tree peony is an important woody ornamental plant that originated in China and has undergone more than 1500 years of artificial cultivation to prolong its flowering period and achieve multi-season flowering. However, there are still some obstacles to achieving tree peony annual flowering under current technologies, such as poor flowering quality and severe cultivar dependence. ‘Qiu Fa No. 1’ is one of our preferred tree peony cultivars for forcing culturing. This cultivar can reflower in late September in north China, but earlier reflowering is still difficult to achieve [2].

GA is an important plant hormone that participates in regulating a variety of growth and development processes, especially flower induction and floral organ development. In woody fruit trees, flower bud differentiation is significantly inhibited when sprayed with GA during the flower induction period [25,26]. In LA lily, GA treatment breaks the bulb’s dormancy and significantly accelerated the flowering process [27]. In rose, spraying different concentrations of GA promotes plant height, flower branch length, and flower stalk length to varying degrees, and caused the plants to bloom earlier [28]. In tree peony, applying GA to help relieve the dormancy of flower buds and promote flowering has become a common method for the off-season regulation of pot flower production [3,29]. In this study, the results showed that GA treatment significantly promoted the differentiation and growth of buds. Since the plant may not have enough nutrients for development after GA treatment, we further applied fertilizer with GA, and this combined treatment achieved the reflowering of tree peony 38 d ahead of the control group (Figure 1A and Appendix A). According to the microstructure observation, the application of GA treatment alone induced the differentiation of bract primordium at 15 d (Figure 2A), and the G + N treatment sustained further development to final flowering. These results indicated that GA played a motivating force in flowering induction and that fertilizer supply ensured continuous bud development to flowering. As this is insufficient, our results lack the comparation of nutrient contents in soil between control and G + N treatment, although the initial condition of the soils is exactly the same, which will be studied in detail in our further work.

In higher plants, flower formation is involved in multiple pathways, forming a complex regulatory network. As an important endogenous hormone, GA promotes flowering mainly by relieving the inhibitory effect of some genes. In general, GA promotes the expression of integration genes in the floral pathway, such as *SOC1*, *FT*, and *LFY*, which also inhibit the expression of temperature-sensitive pathway-related genes, of which *SVP* is a typical example [30]. In Arabidopsis, both *FT* and *SOC1* are significantly induced by GA under short-day growth conditions, and histochemical analysis shows that GA promotes the expression of *FT* and *SOC1* in leaves and stem tips. Furthermore, GA application promotes the expression of *LFY* under short-day conditions [31], or upregulates *LFY* through *SOC1*, which in turn promotes the expression of *AP1*, and ultimately promotes flowering. In contrast, some studies have shown that GA induces a regulatory mechanism by promoting the expression of *LFY*, which can reduce the content of active GA and delay the formation of floral organs [32]. In *Sinapis alba*, *SaMADS*, the homologous gene of *SOC1*, begins to express in the shoot tip and gradually increases after 8 h of GA treatment [33]; in ‘Fuji’ apple (*Malus domestica* Borkh.), the expression of *MdSOC1* is induced by GA_3_ application at 27 d [34]. In our study, among the five flowering pathways-related genes, *PsSOC1* was first induced by GA treatment at 15 d, indicating that *PsSOC1* probably played a key integration role in GA-induced bud differentiation and development. In the G + N combination treatment, *PsFT* and *PsLFY* were induced at 30 d, especially the latter, which not only inhibited the inhibitory expression caused by GA in the later period (60 d), but also exceeded the control during the same period. For the morphological changes, GA treatment alone resulted in the abortion of flower buds, which was consistent with the result in many woody fruit trees [25,26]. In terms of transcriptional expression, GA treatment significantly inhibited the expression of *PsLFY*, which was consistent with the inhibition of *PpLFY* by spraying GA during the flowering induction period of peaches [26]; we also found that fertilizer could reverse GA’s inhibition of *PsLFY* expression. All these results indicated that *PsLFY* played an important role in the response to fertilizer treatment at the late stage of flower bud development. Interestingly, at 45 and 60 d, we found that G + N contrasted with GA treatment in *PsFT* and *PsAP1* expression. This may be due to *AP1* as a downstream gene of *FT*, whose expression was affected by *FT* [14], whereas the different developmental stage of buds between them may also be as one of the reasons for their contrasted expression.

In plants, GA regulates growth and development through biosynthesis and signal transduction. In Arabidopsis, *GA20ox* and *GA3ox* promote flowering, whereas external application of GA downregulates their expression. In apple, during the induction of flower bud differentiation, the expression levels of all four *MdGA2ox* genes are upregulated after GA treatment [35]. In LA lily, the expression of *GA2ox* is significantly induced after spraying exogenous GA, indicating that *GA2ox* has a positive feedback on GA [27]. In *Brassica napus*, the expression of *BnGID1a* is downregulated by GA treatment, showing that *BnGID1a* may be negatively regulated by GA [36]. In Arabidopsis, *GAI*, as one of the *DELLA* genes, mediates the transduction of GA signaling and acts as a negative regulator of GA signaling [37]. To better understand the action mode and regulation mechanism of GA signaling, this study also analyzed the expression of five genes during the process of flower bud differentiation and growth in tree peony. We found that fertilizer reverse the decreasing expression of both *PsGA20ox* and *PsGA3ox* by GA treatment. Combined with the results of GA_3_ content and *PsGA2ox* expression at 15 d, it could be inferred that GA treatment induced the production of endogenous GA, whereas high expression of *PsGA2ox* was initiated to maintain the balance of bioactive gibberellin. These results indicated that *PsGA2ox* may be mainly involved in GA regulation. In addition, GA treatment reduced the expression of both *PsGA20ox* and *PsGA3ox* at 15 d and 30 d, which may be viewed as negative feedback. Since the regulation of GA biosynthesis showed us a huge and complete network, some other factors may also be involved in this process, although more direct evidences is still needed. The rapid decrease in the expression of *PsGAI* caused by GA treatment at 15 d was delayed in the G + N group, indicating that *PsGAI* may regulate further flower formation after fertilizer application. The expression of *PsGID1c* was significantly downregulated after 30 d in both GA treatment and G + N groups, which may have a certain relationship with the induction of *PsGA2ox* and GA_3_ content with negative feedback. By comparing the gene expression between floral pathway and GA signal pathway, we found that the over-expression of most floral pathway-related genes was later than that of GA signaling-related genes, which might be due to the GA signaling-related genes being on the upstream of the floral pathway related genes according to the flower induction network regulation model [29].

The sugar signal pathway, especially the TP6-SnRK1 interaction model, has an important regulatory role in stress responses and plant life activities [38]. T6P, as both a nutrient substance and a signaling molecule, regulates the formation of flowers, whose biosynthesis is regulated by TPS1 [7]. In Arabidopsis, the high expression of *TPS1* increases the content of T6P and promoted flower formation [39]; in addition, the *TPS1* also impacts the vegetative phase change in Arabidopsis, and the loss-of-function of *TPS1* mutant shows a significant delay in the vegetative phase [40]. In our study, G + N treatment, but not GA treatment alone, inhibited the expression of *PsTPS1* at 15 and 30 d, which may give the bud more time to remain in vegetative phase for nutrient accumulation before flowering. Nevertheless, we speculated that fertilizer may play a key role in sugar signaling induction for tree peony reflowering in autumn.

SnRK1 plays an important role in nutrient perception and stress response. It can reduce metabolic activities under adverse conditions and improve environmental adaptability. At the same time, SnRK1 also participates in the coordination of carbohydrates between the ‘source’ and ‘sink’ to ensure the reasonable distribution and utilization of nutrients. Furthermore, SnRK1 is generally considered to be a negative factor regulating growth [41]. In tomato, overexpression of the *SnRK1* gene from apple results in the hindrance of carbohydrate utilization, and the growth of the main shoots is obviously inhibited [22]. In peach, to explore the regulatory role of *PpSnRK1* during the flowering process, the gene *PpSnRK1* overexpression caused a significant delay in the flowering period [42]. In Arabidopsis, silencing the *SnRK1.1* and *SnRK1.2* genes results in early flowering of the plant [23]. In this study, GA treatment fast-induced a high expression of *PsSnRK1* at 15 d, whereas G + N had no such effect, This result indicates GA may cause nutrient deficiency during the triggering of bud sprouting, which may damage its further development. The sugar content at 15 d indicated that GA treatment consumed sucrose but did not increase the content of glucose and fructose, resulting in nutrient deficiency. In response to the stress, it induced *PsSnRK1* expression and reduced the nutrient demand. In G + N treatment, the fertilizer provided more nutrients, and as a signal, the expression of *PsSnRK1* decreased.

Studies have shown that HXK1, as a sugar transport signal, can inhibit plant growth. In transgenic tomato, overexpression of *AtHXK1* significantly inhibits plant growth and development and causes plant senescence and delays flowering [43]. In tree peony, the expression of *PsHXK1* is inhibited by defoliation and GA treatment and induced flowering during the progression of spring festival forcing culture [3]. In this study, *PsHXK1* responded rapidly to the exogenous GA, and then it maintained at a relative higher level up to 45 d, which further confirmed the nutrient deficiency after GA treatment. In short, a sufficient nutrient supply is necessary to further trigger tree peony reflowering at an earlier date in autumn under current forcing culture technologies (Figure 8).

## 4. Materials and Methods

### 4.1. Plant Materials and Treatments

This study was conducted from end of May (one week after the petals withered) to mid-August, 2020. A total of 30 healthy, five-year-old tree peony plants ‘Qiu Fa No. 1’ with similar growth vigor that were free of pests and disease were obtained from the Peony Germplasm Resources Nursery of the Institute of Vegetables and Flowers, Chinese Academy of Agricultural Sciences, Beijing, China (116°15′51″ N, 40°33′32″ E, at 673 m a.s.l.). The plants were randomly divided into three groups for the forcing culture treatments and the first day of these treatments was set as 0 d.

(1) Control group: The plants only received routine maintenance, without any treatment before the forcing culture process.

(2) GA treatment group (G): The leaves of whole plant were sprayed with 200 mg L^−1^ gibberellic acid (GA_3_, Sigma-Aldrich, Shanghai, China) between 8:30 am and 9:30 am. The treatment was performed once a week, four times in total. Other processes were the same as the control group.

(3) GA + nutrient (G + N) treatment group: The whole leaves were sprayed with 200 mg L^−1^ gibberellic acid and 2 g L^−1^ water soluble fertilizer between 8:30 am and 9:30 am. The selection of fertilizer was based on our previous results with N:P:K = 1:3:2 (Appendix A), Mg and other microelements were obtained from the Scotts Company (Marysville, OH, USA). The other processes were the same as the GA group.

On average, an individual plant has 8 branches, each branch has 9 leaves, and each leaf area is about 370 cm^2^. For all three groups, we further treated them with autumn-flowering treatment from 75 d [4] for flowering observation.

### 4.2. Paraffin Section and Scanning Electron Microscope Observations

In this study, flower buds sampled at 0, 30, and 60 d were fixed in FAA fixative for 48 h, referring our previous paraffin section operation instructions [2]. Permanent microscope slides were made and observed under an upright microscope (DM5500, Leica, Wetzlar, Germany).

Based on the results of morphological observations, the flower buds sampled at 0, 30, and 60 d fixed in 2.5% glutaraldehyde fixative solution (with 1% osmium acid) were selected for further scanning electron microscope (SEM) observation. The temporary slices were made according to the SEM sample preparation protocols [4], and starch grains at the base of flower buds were observed with SEM (Cold Field-Emission Scanning Electron Microscope, SU-8010, Hitachi Corporation, Tokyo, Japan).

### 4.3. Soluble Sugar and GA_3_ Content Measurement

Soluble sugars content, including sucrose, glucose and fructose, were determined according to the method of [4], using a DIONEX ICS-3000 ion chromatography system (Thermo Fisher, Carlsbad, CA, USA). The analysis column was a CArboPac PA10 (4 × 250 mm) column, and an electrochemical detector with pulse ampere detection mode was used.

GAs contents were detected by MetWare (http://www.metware.cn/ (accessed on 5 November 2020)) based on the AB Sciex QTRAP 6500 LC-MS/MS platform.

### 4.4. Total RNA Extraction and cDNA Synthesis

The total RNA was isolated using an RNA Prep Pure Kit for plants (Aidlab, Beijing, China), following the manufacturer’s instructions. RNA quality and quantity were assessed by 1.0% agarose electrophoresis and a NanoDrop 2000c spectrophotometer (Thermo Scientific, Waltham, MA, USA), respectively. The first-strand cDNA synthesis was performed using a FastQuant RT kit (With gDNase, Tiangen, Beijing, China) according to the operating instructions.

### 4.5. RT-qPCR

The primers were designed using PrimerPremier 5.0 software and synthesized by Sangon Biotech (Beijing, China). The genes were divided into three categories: (1) the genes in the floral pathway, including *PsFT*, *PsSOC1*, *PsLFY*, *PsAP1*, and *PsSVP*; (2) the genes in the GA pathway, including *PsGA20ox*, *PsGA3ox*, *PsGA2ox*, *PsGAI*, and *PsGID1c*; (3) the genes related to the sugar signal pathway, including *PsTPS1*, *PsSnRK1*, and *PsHXK1*. The primer and gene information were based on Lv et al. [44] and Sun et al. [45] and detailed in Appendix A.

The relative expression levels of different genes were calculated using a double standard curve according to the CFX96 Real-Time system (Bio-Rad, Hercules, CA, USA), and the PCR program was as follows: an initial stage of 30 s at 95 °C; 40 cycles of 5 s at 95 °C, 30 s at 60 °C, and 10 s at 95 °C; *Actin* was used as the reference gene according to Zhang et al. [46], and the 2^−ΔΔCt^ method was used for relative gene expression data analysis [47].

### 4.6. Data Analysis

IBM SPSS Statistics 22 and Origin 9.0 software were used for the experimental data analysis. One-way ANOVA with post hoc Tukey’s test was used for the gene expression level analysis, with *p*-values < 0.05 considered to be significant.

## 5. Conclusions

In this study, G + N treatment shifted the tree peony reflowering time 38 d ahead in autumn, through forcing culture technologies. In this process, GA triggered the flowering induction, and fertilizer ensured a continuous nutrient supply for further bud development. With regarding to transcriptional levels, *PsSOC1* and *PsLFY* played key integration roles in bud differentiation induced by GA and nutrient supply by fertilizer application, respectively. *PsGA2ox* was mainly involved in the GA signal pathway, whereas *PsGAI* may regulate further flower formation after fertilizer application. The sugar signal pathway was also involved in this process, and a sufficient nutrition supply was necessary for earlier GA-induced tree peony reflowering in autumn. Our work may provide a new reference for improving current tree peony forcing culture technologies to achieve its annual flowering with better quality.

## Figures and Tables

**Figure 1 ijms-23-07703-f001:**
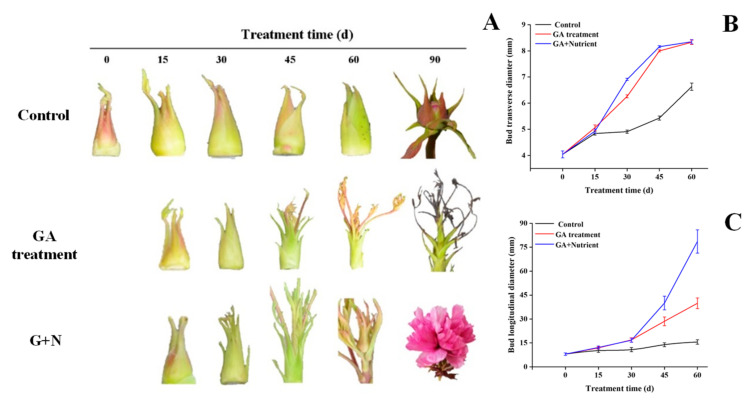
Effects of different treatments on the appearance and morphology of tree peony ‘Qiu Fa No. 1’ flower buds. (**A**) Morphological changes. (**B**,**C**) Variation trend of flower bud’s horizontal and longitudinal diameter. Error bars indicate the SD (*n* = 10).

**Figure 2 ijms-23-07703-f002:**
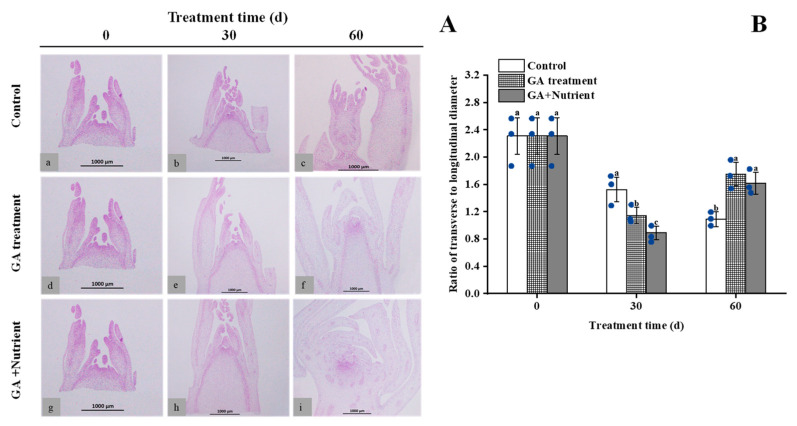
Effects of different treatments on the microstructures of tree peony ‘Qiu Fa No. 1’ flower buds during growth and development. (**A**) Development period of the internal microstructure of flower bud (scale bar = 1000 μm): a, b, d, g: Undifferentiated stage; c, e, h: bract primordium stage; f: sepal primordium stage; i: pistil primordium stage. (**B**) Histogram of flower bud microstructure ratio of horizontal and longitudinal diameters, error bars indicate the standard deviation (*n* = 3), and different lowercase letters indicate significant differences (Tukey’s test, α = 0.05). The blue dots indicate the actual data.

**Figure 3 ijms-23-07703-f003:**
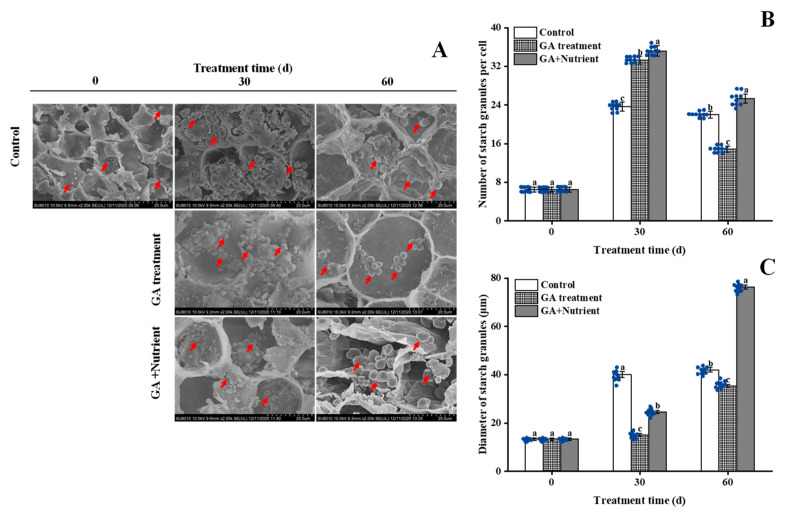
Effects of different treatments on the changes in starch grains at the base of tree peony ‘Qiu Fa No. 1’ flower buds during growth and development. (**A**) Electron microscope micrographs of starch grains at the base of flower buds in the control and different treatment groups (red arrow indicates the starch grains), magnified 2000 times (scale bar = 20 μm). (**B**,**C**) Histogram of the number and diameter of starch granules in a single cell, respectively. Error bars indicate the SD, in each treatment group, five individual plants were used for starch granule number and size analysis with two samples from the same plant (*n* = 10), and different lowercase letters indicate significant differences (Tukey’s test, α = 0.05). The blue dots indicate the actual data.

**Figure 4 ijms-23-07703-f004:**
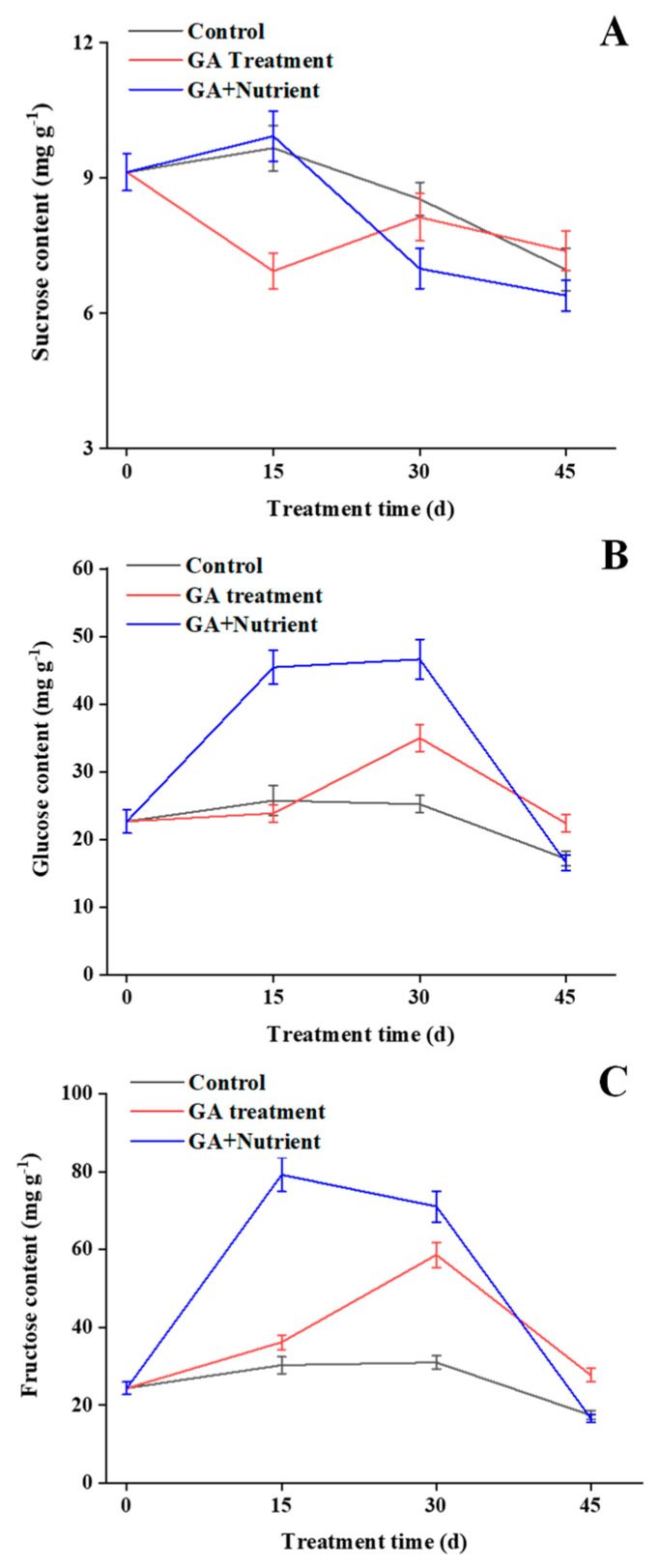
Effects of different treatments on soluble sugar content in buds of tree peony ‘Qiu Fa No. 1’. The content of (**A**) sucrose, (**B**) glucose, and (**C**) fructose during the period of 0–45 d. Error bars indicate the standard deviation (SD; *n* = 3).

**Figure 5 ijms-23-07703-f005:**
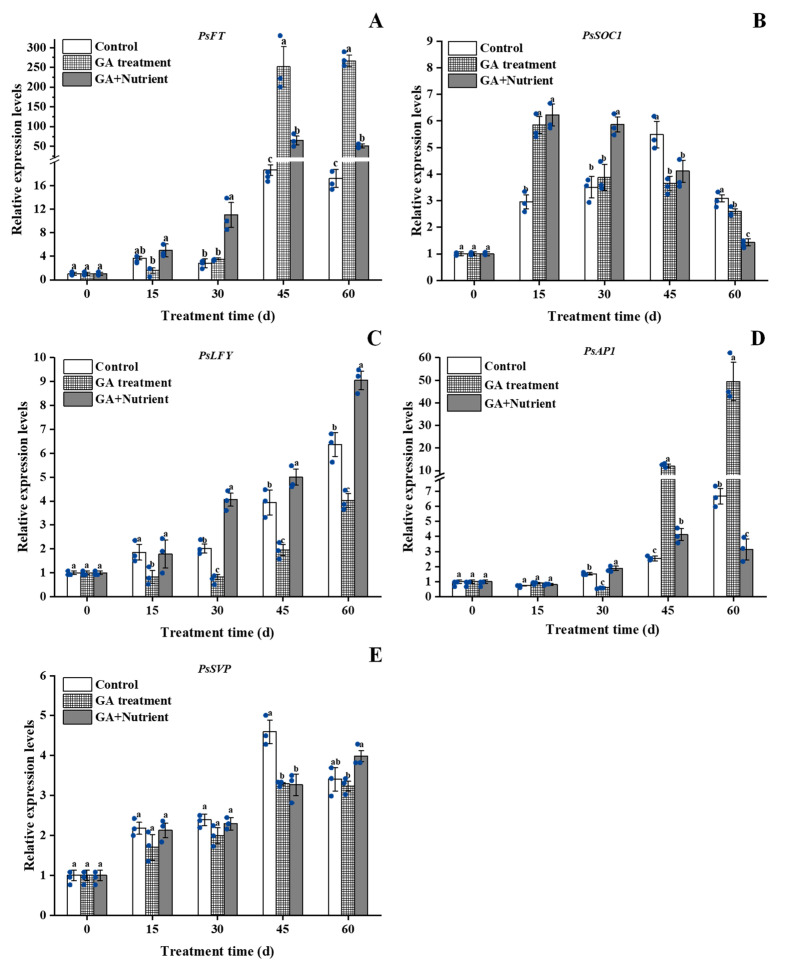
Effect of GA treatment and GA + Nutrient treatment on the expression of (**A**) *PsFT*, (**B**) *PsSOC1*, (**C**) *PsLFY*, (**D**) *PsAP1*, and (**E**) *PsSVP* in the buds of tree peony ‘Qiu Fa No. 1’. Error bars indicate the SD (*n* = 3). Different lowercase letters indicate significant differences (Tukey’s test, α = 0.05), and the blue dots indicate the actual data.

**Figure 6 ijms-23-07703-f006:**
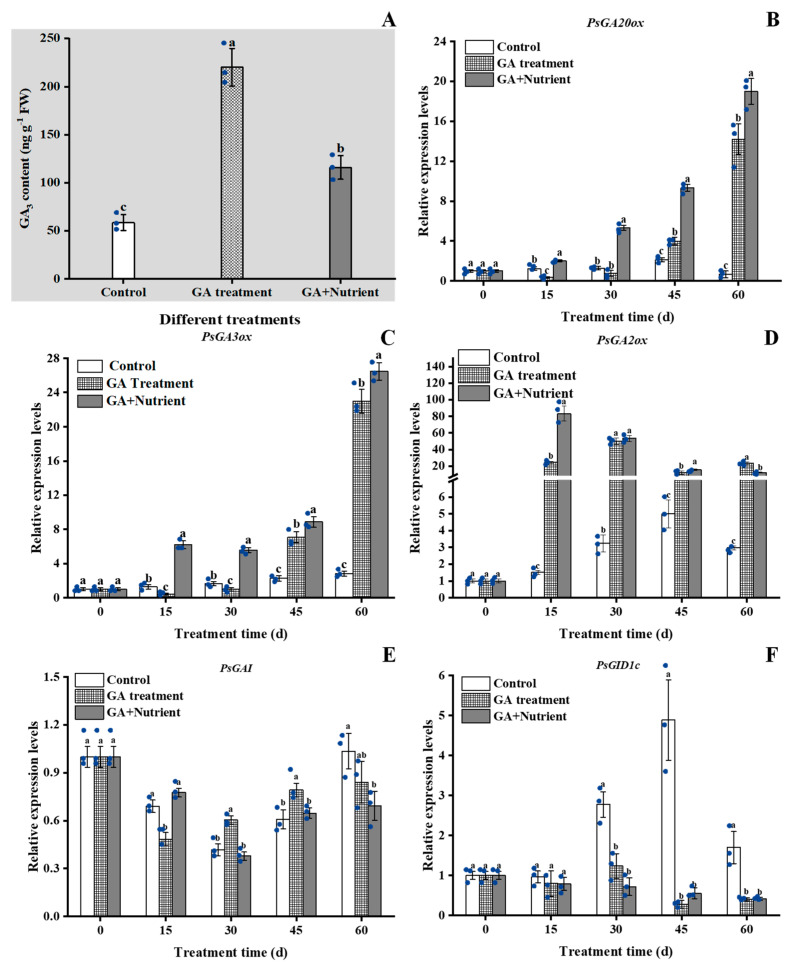
Effect of GA treatment and GA + Nutrient treatment on the (**A**) GA_3_ content (in the shaded area) and expression of (**B**) *PsGA20ox*, (**C**) *PsGA3ox*, (**D**) *PsGA2ox*, (**E**) *PsGAI*, and (**F**) *PsGID1c* in the buds of tree peony ‘Qiu Fa No. 1’. Error bars indicate the SD (*n* = 3). Different lowercase letters indicate significant differences (Tukey’s test, α = 0.05), and the blue dots indicate the actual data.

**Figure 7 ijms-23-07703-f007:**
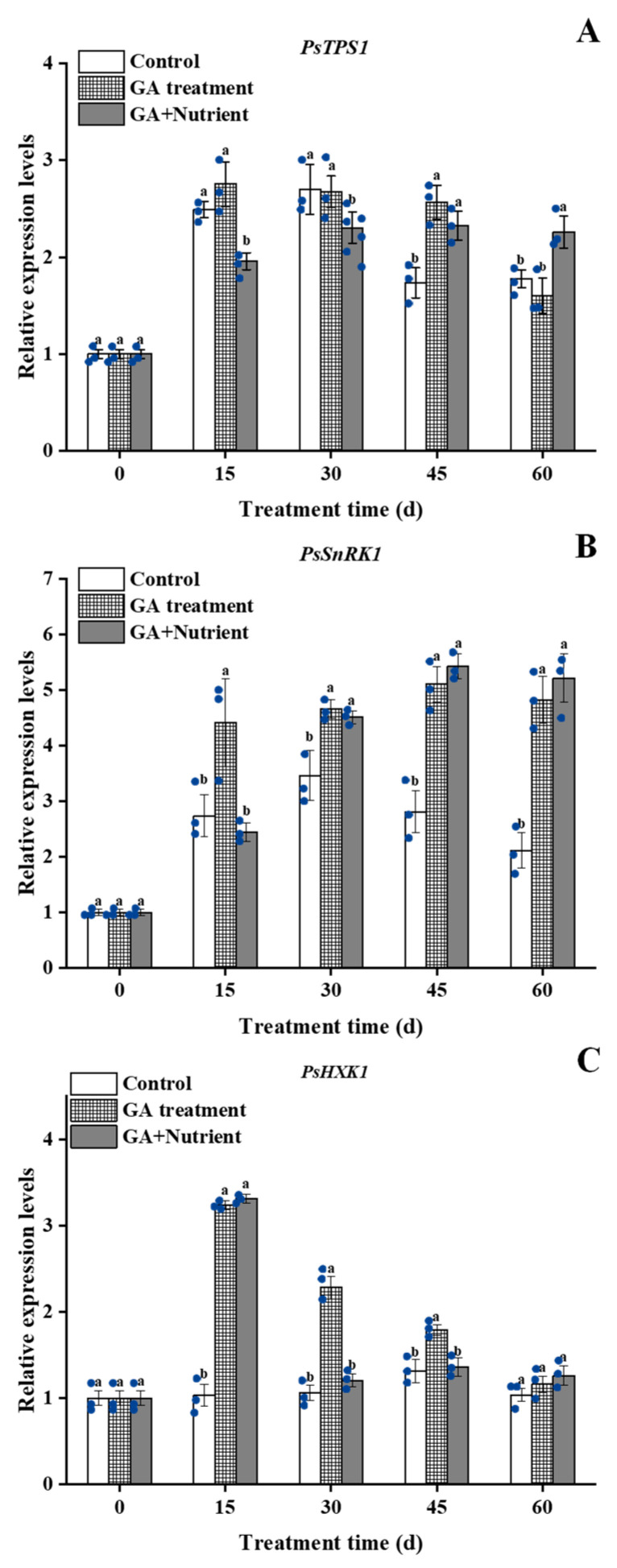
Effect of GA treatment and GA + Nutrient treatment on the expression of (**A**) *PsTPS1*, (**B**) *PsSnRK1*, and (**C**) *PsHXK1* in the buds of tree peony ‘Qiu Fa No. 1’. Error bars indicate the SD (*n* = 3). Different lowercase letters indicate significant differences (Tukey’s test, α = 0.05), and the blue dots indicate the actual data.

**Figure 8 ijms-23-07703-f008:**
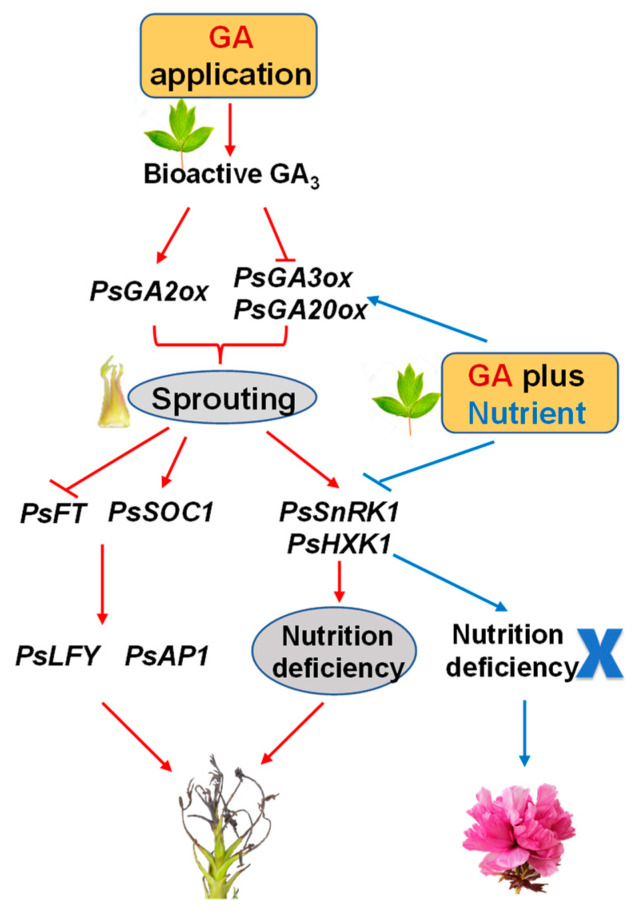
Schematic diagram of regulation mode in gibberellin induced tree peony reflowering in autumn. GA treatment induced reflowering, but led to abortion of the bud finally (red line), probably due to nutritional deficiency. Adding of fertilizer supplied sufficient nutrition (blue line), thus the flower reflowered normally. The initial (15 d or 30 d) inducing or decreasing expression of related genes are signed by the arrows and lines with a bar at the end, respectively.

## Data Availability

The data used to support the findings of this study are available from the corresponding authors upon request.

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
