# Peer review of "Nutrient Supply Is Essential for Shifting Tree Peony Reflowering Ahead in Autumn and Sugar Signaling Is Involved"

_ijms, 2022, doi:10.3390/ijms23147703_

Round 1
Reviewer 1 Report
Th findings presented in this work are potentially very interesting. The authors discovered that gibberellin (GA) and fertilizer treatment significantly induced the expression levels of flower-specific genes, affected the sugar signaling and induced earlier flowering in tree peony. Although I think the authors make here an important contribution, I have a number of comments concerning certain statements and conclusions which, in my opinion, should be addressed in order to improve the manuscript:
Major comments:
- I would suggest the authors to clarify the title of the manuscript to make it more focused.
- In my opinion, the abstract requires a number of clarifications. The authors use terms as PsTPS1, PsSnRK1, but don’t mention what are they and what is their role. Part of the abstract reads as a part of results section describing specific experiments and not summarizing the main findings. At the same time, the use of “fertilizer” term here is very vague and general without mentioning anything about its composition or choice which is very relevant here.
- In general, the text contains a lot of style mistakes which have to be revisited, corrected and clarified. A good English check is necessary.
- Line 32. What is “a sufficient nutrition supply”? what is it depending on? What is the meaning of “forcing culture technology”? Correct the sentence and clarify the statements.
- Line 54. It would be important in this paragraph to shortly specify in which plants certain pathways have been identified and if they are conserved among all flowering plants. Currently, the description is very generalized.
- Line 76. The GA pathway is not properly described here.
- Line 95. Indicate properly the reasoning for why “Qiu Fa No. 1’ was used here.
- Line 96. The term “fertilizer” appears here without any clarification why this particular fertilizer was used, what is so special about its composition etc. This is important and a better description and clarification is needed.
- Figure 3. How is the cell size varying in the same bud? Is it different at the edges and its center? It looks like the cell size increases after GA treatment, but then it looks like its smaller again in the combined treatment. Can the authors comment on that?
At the same time, I think it’s not necessary to copy three times same picture of the control.
- It is not very clear how Qiu Fa No. 1’ is integrated in the figures and this study in general.
- Line 426. “GA may have promoted flowering induction”- why in some cases the authors doubt the main conclusions and in other cases make clear statements? This is very confusing, as it makes an impression of lack of confidence in certain results and data. Please clarify.
Minor comments:
- Line 20. “….completed the opening process 38 d before the control group….” As this is the Abstract, basic abbreviations should be avoided and 38 d should be indicated as 38 days.
- Line 21. Do the authors mean positive regulation when using “motivating force” phrasing? Please check through ought the text and correct the style mistakes.
- Line 37. “highly influential woody plant” sounds confusing. Correct the style throughout the text.
- Line 61 and 64. The references are missing. Which plant species were used?
- Line 70. GA is hardly just a “endogenous signal participant”, correct the sentence.
- Line 88. What are these “most plants”?
- Line 213. “Micrographs” instead of “Microstructures”.
- Line 224. What is “ the greater level”?
- Line 300. What does “forcing culturing” Mean?
Reviewer 2 Report
The manuscript describes how to make earlier the flowering in tree peony. The authors' finding is a combination of gibberellin (GA) and fertilizer treatments successfully shortened the period for the flowering. The authors also checked the expression of some flowering, GA biosynthesis/metabolism, and sugar signalling-related genes to discuss how the GA and fertilizer treatment promote the flowering earlier in tree peony. Comments and questions are as follows.
1. Total English proofreading is required because some sentences are unclear, and there are typos; e.g. the title, "sugar signaling is involved" for what?; the first sentence in the abstract "Tree peony has restricted florescence," seems to be a misspelling, but still strange if it means flowering time.
2. Line 113 in the methods. "The whole leaves were sprayed with 200 mg L-1 gibberellic acid".
Please add the information of GA types and company purchased.
3. The nutrient information about soil is lacking. It is helpful if the authors add information about the contents of N, P, K in the soil and how the fertilizer treatment increased these nutrients because this is related to the authors' main claim.
3. Lines 152-154 in the methods. "(1) the genes in the floral pathway, including PsFT, PsSOC1, PsLFY, PsAP1, and PsSVP; (2) the genes in the GA pathway, including PsGA20ox, PsGA3ox, PsGA2ox, PsGAI, and PsGID1c; and (3) the genes related to the sugar signal pathway, including PsTPS1, PsSnRK1, and PsHXK1."
Please add the Genbank accession numbers of these listed genes.
4. Figure 1. There is no information about how many plants successfully developed flowers. The information on flowering plant ratio is required.
5. Figure 3. The authors examined starch grains with the electron microscope and calculated their number and diameter. The data are the sum of 10 samples, but it is unclear whether the samples came from one plant or several plants. Because there will be an individual plant variation, the analysis should be done with several plants within a treatment.
6. Lines 314-315. "we further applied fertilizer with GA, and the combination treatment achieved the reflowering of tree peony 38 d earlier than that of our previous reports (Wang et al., 2020)".
It seems two experiments (this work and Wang et al., 2020) were conducted in different years. Therefore it will be difficult to compare the two experiments directly to conclude 38 days earlier flowering.
7. Lines 334-337. "In Sinapis alba, SaMADS, the homologous gene of SOC1, begins to express in the shoot tip and gradually increases after 8 h of GA treatment (Bonhomme et al., 2000). In our study, among the five flowering pathways-related genes, PsSOC1 was first induced by GA treatment at 15 d, indicating that PsSOC1 probably played a key integration role in GA-induced bud differentiation and development."
The conclusion is not convincing because the time scale is very different between the two experiments.
8. Lines 367-369. "PsGA2ox expression at 15d, it could be inferred that GA treatment induced the production of endogenous GA, whereas high expression of PsGA2ox was initiated to maintain the balance of bioactive gibberellin."
It is an unclear statement because GA solely treatment reduced PsGA3ox and PsGA20ox (Figures 6B and C).
9. Lines 386-388. "We speculated that the application of fertilizer promoted the increase of T6P content, and as a feedback and response to the suppression of PsTPS1, which also indicated that fertilizer may play a key role in sugar signaling induction for tree peony reflowering."
This is unconvincing because TPS1 and T6P synthesis promote flowering (Wahl et al.,
Science. 339: 704-707), whereas PsTPS1 expression does not support this.
10. Figure 8. The diagram is unclear; what is the difference between red and blue liens. Ps FT is suppressed, although it is required for flowering. The regulatory network does not mirror the data (e.g. PsSnRK expressions are almost the same between GA and GA+firtilizer at 30 and 45 days, PsHXK1 expression is also at 15 day).
Reviewer 3 Report
Dear Authors,
Your peony report from IJM was extended for my peer review. Please find below suggestions for improvement. Important issues are flagged by line number, and crucial ones with [MAJOR]. I feel this paper has merit, but I do not feel comfortable recommending its publication just yet.
16 "restricted florescence" Imprecise. What is restricted? Time? Size? Number of flowers? Other aspects?
18 fertilizer -> {for symmetry, introduce (G+F) here}
22 energy -> {Throughout, please change to "nutrients". There seems to be no direct energy source (sugars?) in the fertilizer used}
24 integration -> integral
30 vs. 31: insufficient nutrition vs. energy shortage {That's not the same thing!!!}
37 highly influential
38 attracted by -> attracting
[MAJOR] Missing information on economic importance of peony. Can use volume of trade ($ value in China, or global), number of registered/released cultivars, etc.
52-53 {Sentence "In these processes..." is a loop logic. Please rewrite or delete}
54 At least six floral induction pathways have been reported {What is meant? In what aspects were they reported?}
70 such -> including
82 substance -> compound (or: molecule). {Because later several are brought up, use in plural: compounds (or: molecules), respectively}
108 673 m {if that's altitude, append with "a s l"}
110 (GA) and (G+F) not defined yet.
115 {Add approximate volume per leaf (or tree), respectively}.
150-160 [MAJOR] How EXACTLY were the primers constructed? - Provide the genome annotation details used. For the genes analyzed, provide the Ps(geneIDs) or NCBI sequence IDs (this also is needed for actin). Here, you describe the 2-DDCT, but in figures or text it's not clear what the baseline is; further, the figures with expression data to not identify the baseline either.
164 [MAJOR] What is described should rather use ALPHA = 0.05 not P < 0.05; at this given ALPHA different numbers will denote significant differences. Correct throughout, including all figure legends that apply.
Figure 2, 3, 5, 6, 7. [MAJOR] Replace the bar graphs with the actual data. Use beeswarms or dot plots, with additional markings for mean/SD or median / +- 25%.
202 results -> study
299 previous
314 the combination -> G+F {or: this combined}
316 (application of) GA treatment
317 with no fertilizer application
349 antagonized -> contrasted; This may (be)
350 affected by FT {support with citation(s)}
352 antagonized -> contrasted
354 while -> whereas
356 expressions -> expression levels
358 passivation {Meaning unclear; please rewrite)
374 inducing -> induction
380 In recent years, it has been reported that
382 activities {Supported with citations, as this seems to have been reported}
398 is transferred to Arabidopsis and
399 overexpression (in Arabidopsis) causes caused
409-410 {What aspect of HXK1 indicates energy shortage?}
415 induction of
416 energy -> nutrient
417 shift -> trigger
433 [MAJOR] Your conclusions is a repetition (AGAIN, after re-appearance in Discussion) of the major results. In this section, instead show a broader implication of your findings. And, what future research/ specific applications does your study enable?
Round 2
Reviewer 2 Report
The manuscript is improved, but the title still has a misspelling.